# Comprehensive Similarity Algorithm and Molecular Dynamics Simulation-Assisted Terahertz Spectroscopy for Intelligent Matching Identification of Quorum Signal Molecules (N-Acyl-Homoserine Lactones)

**DOI:** 10.3390/ijms25031901

**Published:** 2024-02-05

**Authors:** Lintong Zhang, Xiangzeng Kong, Fangfang Qu, Linjie Chen, Jinglin Li, Yilun Jiang, Chuxin Wang, Wenqing Zhang, Qiuhua Yang, Dapeng Ye

**Affiliations:** 1Center for Artificial Intelligence in Agriculture, Fujian Agriculture and Forestry University, Fuzhou 350002, China; zlt@fafu.edu.cn (L.Z.); clj@fafu.edu.cn (L.C.); yilun@fafu.edu.cn (Y.J.); chuxin@fafu.edu.cn (C.W.); zwq@fafu.edu.cn (W.Z.); ydp@fafu.edu.cn (D.Y.); 2Fujian Provincial Key Laboratory of Terahertz Functional Devices and Intelligent Sensing, Fuzhou University, Fuzhou 350108, China; jimlinlee@fzu.edu.cn; 3Fisheries Research Institute of Fujian, Fuzhou 350025, China; qhyang1314@163.com

**Keywords:** Gram-negative pathogens, quorum signaling molecules, terahertz spectroscopy, molecular dynamics simulation, spectral similarity calculation

## Abstract

To investigate the mechanism of aquatic pathogens in quorum sensing (QS) and decode the signal transmission of aquatic Gram-negative pathogens, this paper proposes a novel method for the intelligent matching identification of eight quorum signaling molecules (N-acyl-homoserine lactones, AHLs) with similar molecular structures, using terahertz (THz) spectroscopy combined with molecular dynamics simulation and spectral similarity calculation. The THz fingerprint absorption spectral peaks of the eight AHLs were identified, attributed, and resolved using the density functional theory (DFT) for molecular dynamics simulation. To reduce the computational complexity of matching recognition, spectra with high peak matching values with the target were preliminarily selected, based on the peak position features of AHL samples. A comprehensive similarity calculation (CSC) method using a weighted improved Jaccard similarity algorithm (IJS) and discrete Fréchet distance algorithm (DFD) is proposed to calculate the similarity between the selected spectra and the targets, as well as to return the matching result with the highest accuracy. The results show that all AHL molecular types can be correctly identified, and the average quantization accuracy of CSC is 98.48%. This study provides a theoretical and data-supported foundation for the identification of AHLs, based on THz spectroscopy, and offers a new method for the high-throughput and automatic identification of AHLs.

## 1. Introduction

The deterioration of the aquatic environment and the development of intensive high-density aquaculture models have resulted in the increased occurrence of aquatic bacterial diseases [1]. The sustainable development of modern aquaculture is significantly constrained by bacterial diseases. Antibiotics are generally acknowledged as effective for preventing and controlling bacterial diseases in aquaculture, but their misuse increases the risk of residuals and worsens the dissemination of resistance genes [2,3,4]. It has been found that many aquatic pathogens coordinate the expression of virulence factors by secreting, accumulating, and sensing signaling molecules based on population density variations through a mechanism known as quorum sensing (QS) [5,6]. Thus, the mechanism of quorum quenching (QQ) through the enzymatic degradation of signal molecules is considered a new bacterial disease control strategy that has the potential to replace traditional treatments for bacterial diseases in aquaculture [7,8,9]. The major signal molecules for QS in common aquatic pathogens, which are predominantly Gram-negative bacteria, are N-acyl-homoserine lactones (AHLs) [10]. A homoserine lactone ring that is linked to an acyl group is a shared characteristic of AHLs [11]. However, the acyl chain length (*n* = 4~18) and the substituent (hydrogen, hydroxyl, or carbonyl) on the third carbon atom exhibit diversity, which is influenced by variations in the bacterial community and environmental conditions [12]. The regulation of the expression of numerous different virulence factors by diverse AHL signaling molecules presents challenges in preventing and controlling bacterial diseases in aquaculture. The accurate identification of various species of AHLs is of great scientific value and provides a basis for revealing the complex and multiple differential action modes of AHLs in the QS system of Gram-negative pathogens and for assisting in the development of QQ-based strategies for the targeted control of bacterial diseases in aquaculture.

Currently, bacterial biosensors, gas- and liquid-phase-based chromatographic methods, and various sensor methods are the most commonly employed methods for bacterial signal molecular identification. The bacterial biosensor method is a traditional approach to identifying AHLs. The main AHL sensors, such as *Agrobacterium tumefaciens NTL4* [13], *Chromobacterium violaceum CV026* [14], and *Escherichia coli PSB401* [15], possess high sensitivity and specificity, as well as environmental friendliness in AHL identification. However, this method requires tailored strains for AHL identification, which is a time-consuming and laborious process. Gas- and liquid-phase-based chromatographic methods are the most widely used techniques to identify AHLs, including thin-layer chromatography (TLC) [16], gas chromatography-mass spectrometry (GC-MS) [17], liquid chromatography-mass spectrometry (LC-MS) [18], and liquid chromatography-tandem mass spectrometry (LC-MS/MS) [19,20]. Despite exhibiting high identification accuracy and sensitivity, these methods have the limitations of complex pre-processing operations and high technical requirements for skilled inspectors. Consequently, the identification process is time-consuming, laborious, and unsuitable for rapid on-site identification. In addition, various new sensor methods, such as the quartz crystal microbalance (QCM) sensor [21], photoluminescence sensor [22], and fluorescent sensor [23], which incorporate specific recognition elements are emerging endlessly. Although these techniques have greatly enriched the means of AHL identification, there remain challenges in meeting practical application needs. Continuous advances in science and technology have led to the emergence of new molecular fingerprint spectroscopy methods in the form of terahertz (THz) spectroscopy, which provides novel strategies for substance identification [24,25]. THz spectroscopy is a cutting-edge frontier technology offering the outstanding advantages of fingerprinting, coherence, and penetration. Its frequency range lies between the transition regions of macroscopic electronics and microscopic photonics, which can simultaneously detect the rich physicochemical properties of substances [26]. Moreover, with the advancement of ultrashort pulse lasers, this THz identification capability has been further tapped into, greatly promoting the theoretical depth and application breadth of THz detection technology [27,28]. THz spectroscopy has unique fingerprinting advantages in probing molecular spatial structures, intermolecular reactions, and interaction forces, which provide obvious fingerprinting characteristics for identifying molecular conformation and configuration [29,30,31]. This provides a novel approach to identifying AHLs and provides a theoretical basis and data support for establishing a comprehensive THz fingerprint spectra database of AHLs.

The THz fingerprint spectral characteristics of AHLs not only contribute to our understanding of their basic material properties but can also be combined with artificial intelligence algorithms to achieve automatic and rapid identification of AHLs. However, given that AHL molecules share a homoserine lactone ring and similar acyl side chain structures, their spectral characteristics may exhibit high similarity. Consequently, combining spectral similarity identification algorithm strategies offers an effective solution to distinguish between the similar THz fingerprint spectra of similar molecules. The current approaches for similarity recognition mainly consist of distance algorithms, similarity algorithms, and machine learning algorithms. In the THz spectroscopy field, the spectral features utilized by this technique, combined with machine learning algorithms, are the main means to achieve molecular recognition. Tu et al. [32] proposed an improved support vector machine classification model based on variational mode decomposition and particle swarm optimization to achieve the identification of the isomers of organic molecules. Chen et al. [33] combined principal component analysis and fuzzy pattern recognition methods to identify the THz spectra of amino acid and sugar biomolecules. Yi et al. [34] proposed an improved FCM algorithm for the THz spectral recognition method and achieved a clustering accuracy of 0.9668. The above results show that THz spectroscopy, combined with machine learning algorithms, can achieve effective recognition of the different types of biomolecules. However, the training and validation process of machine learning algorithms is cumbersome and requires large data sets. Similarity algorithms and distance algorithms have rarely been reported in the field of THz molecular recognition. Nevertheless, based on the preliminary research explorations in our previous research work, terahertz spectroscopy, when combined with similarity or distance algorithms, is feasible for target molecule recognition [35]. However, it is worth noting that distance algorithms are highly sensitive to outliers, which can easily lead to large deviations in the calculation results. Similarity algorithms have high computational complexity and are costly in terms of the time needed for matching with large databases. Therefore, maximizing the use of THz fingerprint characteristics to reduce the amount of computation and optimizing the algorithm framework to improve the recognition accuracy of the algorithm deserve further research. Hence, there is a pressing need for a similarity-matching algorithm that can provide higher accuracy, greater robustness, and faster results in order to effectively address the challenge of identifying the THz fingerprint spectra of molecules that are similar in nature.

In this paper, we propose a high-throughput and automatic matching recognition method for eight types of AHLs with similar molecular structures, aiming to solve the time-consuming, low-precision, and laborious problems in the existing methods for recognizing similar molecules. The specific work described this paper is as follows: (1) THz time-domain spectroscopy (THz-TDS) is utilized to detect the fingerprint spectral characteristics for eight types of AHLs, including C4-HSL, C6-HSL, C8-HSL, 3-oxo-C8-HSL, C10-HSL, C12-HSL, 3-oxo-C12-HSL, and C14-HSL. (2) The molecular geometric configuration and vibrational modes of the AHLs are calculated using density functional theory (DFT). The fingerprint absorption peaks from the experimental spectra are recognized, attributed, and resolved based on DFT, and a THz fingerprint spectral database of the eight AHLs was established. (3) A comprehensive similarity calculation (CSC) method combines the spectral shape similarity algorithm and peak distance algorithm is proposed to calculate the similarity between selected spectra and the targets at two scales (i.e., spectral shape and feature peak distances). This method first performs initial spectra screening, based on the THz fingerprint peak position frequencies, and selects the spectra that have high peak matching values with the targets from the AHL spectra database. This can avoid using the complex similarity algorithm to match all spectra within the AHL database, saving computational resources and reducing identification time. Then, the similarities between the selected spectra and the targets are calculated and sorted, based on the weighted CSC algorithm. The matching identification method proposed in this study enables automatic and efficient identification of AHLs with high accuracy and robustness. This work provides a data-centered basis for the identification of AHLs based on THz-TDS and delivers crucial theoretical and technical support for achieving the high-precision identification of THz fingerprint spectra of similar molecules.

## 2. Results and Discussion

### 2.1. Molecular Geometric Configuration

The single-molecule geometric configuration of the eight AHLs was tightly optimized using Gaussian 2016 software (Gauss Inc., Wallingford, CT, USA) with the B3LYP/6-31G+(d,p) DFT model. The molecular electrostatic potential (MEP) of these optimized molecular structures was generated with GaussView5.08 software (Gauss Inc., Wallingford, CT, USA), as shown in Figure 1. The vibration results in the Gaussview software show that no imaginary frequencies were present. The results indicated that the B3LYP/6-31G+(d,p) DFT model performed well in optimizing the geometry of molecular structures and obtaining stable molecular conformations. Therefore, the atomic coordinates of stable molecular conformations can be utilized as the input to calculate the vibration modes that caused the resonant frequencies so that the fingerprint peak of the signal molecules can be analyzed. Notably, as shown in Figure 1, all eight of the AHLs contain a homoserine lactone ring, which differs only in acyl side chain length (4 to 14 carbon atoms). Furthermore, the maximum MEP value of all these AHLs appeared around the oxygen atoms (in red). These results show that similar molecular structures result in similar molecular vibration patterns and may exhibit similar fingerprint features at THz frequencies.

### 2.2. Comparison of Experimental and Theoretical Spectra

Based on the optimized single molecular structure, the theoretical absorption spectra of the AHLs were calculated. Figure 2 presents a comparison between the theoretical spectra simulated by the B3LYP/6-31G+(d,p) DFT model and the experimental spectra obtained by THz-TDS. It can be seen that except for a slight frequency shift and the absence of a few absorption peaks, the theoretical and experimental spectra show high spectral similarity in both the absorption peak shape and the frequency position. The main reason for the difference between the theoretical and experimental spectra is that the experimental sample is in the form of solid pellets, while the theoretical DFT simulation is based on a single molecule of AHL. Therefore, crystal resonance, crystal field effects, and intermolecular interactions were ignored in the theoretical simulations [36]. Moreover, the experimental spectra were obtained at a lab temperature of 294 K, while the DFT simulation was conducted at 0 K, disregarding the thermal effect. The larger number of theoretical absorption peaks than experimental absorption peaks may be due to the limitations of THz-TDS instruments, which are incapable of detecting all the absorption peaks that arise from molecular vibration.

### 2.3. Assignment of Absorption Peaks

The visualization function available in GaussView 5.08 software (Gaussian Inc., Wallingford, CT, USA) was used to analyze the formation mechanism of the THz characteristic absorption peak. GaussView software can visualize the dynamic vibration of the molecules; therefore, we can obtain the specific assignment of the molecular vibration at the absorption peaks by visualizing the results. According to the results of the DFT simulation, Table 1 lists the assignment of vibration modes that cause the absorption peaks of all eight AHLs. As shown in Table 1, the vibration assignments at the absorption peaks of the eight AHLs are dominated by the C-C bending vibration. We infer that AHLs are likely to have a primarily carbon chain as the backbone for the overall molecular bending vibration, but the different carbon chain lengths and substituent groups will lead to the inconsistent amplitude of the overall molecular bending vibration of the AHLs, which may lead to subtle vibrational differences between the side chain and the homoserine lactone ring. The peaks of C4-HSL at 0.92 and 1.58 THz were generated by the out-plane bending vibration of C-H and the in-plane bending vibration of C-N, respectively. Five peaks of C6-HSL were found at 0.61, 0.83, 0.97, 1.57, and 1.86 THz. The peaks at 0.61 THz and 1.86 THz were generated by the out-plane bending vibration and in-plane bending vibration of C-H, respectively. The peaks at 0.83 THz and 1.57 THz were generated by the in-plane bending vibration and stretching vibration of C-C, respectively. The peak at 0.97 THz was formed by the combined interaction of the benzene ring respiration vibration and the in-plane bending vibration of C-C. Five peaks of C8-HSL were found at 0.66, 0.87, 1.03, 1.57, and 1.78 THz. The peaks at 0.66 THz and 0.87 THz were generated by the respiration vibration of benzene rings. The peak at 0.87, 1.03, and 1.57 THz were all generated by the in-plane bending vibration of C-C. The peak at 1.78 THz was generated by the out-plane bending vibration of C-H. The peaks of 3-oxo-C8-HSL at 0.72 and 1.58 THz were generated by the in-plane flexural vibration of C-C and C-N-C, respectively. There were three peaks for C10-HSL, of which the peaks at 0.73 THz and 1.04 THz were all generated by the in-plane bending vibration of C-C, and the peak at 1.51 THz was formed by the combined interaction of the in-plane bending vibration of C-H and C-C. The peaks of C12-HSL at 0.92 and 1.69 THz were generated by the in-plane bending vibration of C=N and C-H, respectively. Seven peaks of 3-oxo-C12-HSL were found at 0.68, 0.87, 1.07, 1.25, 1.46, 1.84, and 1.87 THz. The peaks at 0.68, 1.46, and 1.63 THz were all generated by the in-plane bending vibration of C-C. The peaks at 1.07 and 1.84 THz were generated by the in-plane bending vibration of C-H. The peak at 0.87 THz was formed by the combined interaction of the in-plane stretching vibration of C=O and the in-plane bending vibration of C-C. The peak at 1.25 THz was formed by the combined interaction of the in-plane bending vibration of C-N and C-H. Four peaks of C14-HSL were found at 0.98, 1.42, 1.61, and 1.75 THz. The peak at 0.98 THz was formed by the combined interaction of the in-plane bending vibration and the out-plane bending vibration of C-H. The peaks at 1.42 and 1.75 THz were generated by the in-plane bending vibration of C-C. The peak at 1.61 THz was formed by the in-plane stretching vibration of C=O. It can be seen that a molecular dynamics simulation based on DFT can analyze the formation mechanism of the THz fingerprint peaks of AHLs and provide a theoretical basis and data support for the subsequent detection of AHLs, based on the THz fingerprint peaks.

### 2.4. Identification of AHLs Based on Traditional Spectral Similarity Algorithm

For the investigation of the spectral characteristics of AHLs that share similar molecular structures, we compared spectral shape similarity algorithms (cosine similarity (CS), cosine-cross similarity (CCS), and Jaccard similarity (JS)) and spectral distance algorithms (Euclidean distance (ED), Hausdorff distance (HD), and discrete Fréchet distance (DFD)) to calculate the THz spectral similarity of the eight AHLs. The computational results are presented in Figure 3. The CS curves (Figure 3a) highlight the limited spectral resemblance between C10-HSL and the rest of the AHLs. The CCS heatmap (Figure 3b) demonstrates the similarity values between each type of AHL and indicates potential mismatches between C4-HSL and 3-oxo-C8-HSL, C6-HSL and 3-oxo-C12-HSL, and C12-HSL and C14-HSL. The JS heatmap (Figure 3c) indicates that the spectral similarities of the same AHLs are all above 0.95, while the similarities between different AHLs are low, demonstrating the effectiveness of spectral shape features for identifying AHLs. The HD curves (Figure 3d) reveal similar spectral distances between C6-HSL, C8-HSL, and C12-HSL, and between 3-oxo-C8-HSL and 3-oxo-C12-HSL. The ED heatmap (Figure 3e) and DFD heatmap (Figure 3f) depict the distances that were calculated based on the full spectra and feature points, respectively. Both indicate that the spectral distances between the same AHLs are small, while the distances between different AHLs are large, demonstrating the effectiveness of spectral distance features for identifying AHLs. With the exception of C10-HSL, which possesses unique THz spectral features, C4-HSL, C8-HSL and 3-oxo-C8-HSL, C6-HSL, and 3-oxo-C12-HSL, as well as C12-HSL and C14-HSL, possess comparable spectral characteristics, all of which are prone to misidentification. To identify the spectral shape similarity and spectral distance more visually, as shown in Figure 3g, the statistical bar above the split line shows the normalized sum of spectral shape similarity (CS, CCS, and JS), while that below the split line shows the normalized sum of spectral distance (HD, ED, and DFD). The higher spectral shape similarity and comparatively lower spectral distance among the AHLs indicate the possibility of effectively distinguishing each AHL. Although the features of spectral shape and distance are effective in identifying AHLs, it is difficult to guarantee the accuracy and stability of matching by relying solely on one feature for matching identification. Additionally, matching identifiers one by one in large databases can be exceedingly laborious. Therefore, a method that is capable of utilizing THz spectral features at multi-scales is needed to match the identification of similar spectra.

### 2.5. Identification of AHLs, Based on the Proposed Two-Step Matching Method

(1)Initial screening of AHLs, based on peak position similarity

The database comprises the average THz absorption spectra obtained from eight replications of the eight AHLs. Although this database is relatively small, it is crucial to reduce the number of matched spectra for complex similarity calculations with the target spectra as the size of the database increases. In this paper, the fingerprint peak position feature of the THz spectra is utilized for the initial screening of spectra in the database, as shown in Figure 4. As depicted in the heatmap of fingerprint peak position matching (Figure 4a), the C4-HSL sample exhibits a high degree of similarity with the 3-oxo-C8-HSL fingerprint peak position. Conversely, the C6-HSL sample exhibits low peak position matching, with a maximum value of 0.6, probably due to the inaccurate spectral acquisition that caused the peak position shifts. Based on the initial screening procedure, we designed a heat map of fingerprint peak position matching, based on peak matching degree Equation (2) (Figure 4a). For all types of AHLs, 2 to 4 spectra were selected for further comprehensive similarity calculations. Therefore, according to the screening program, the number of spectra for subsequent similarity matching is appropriate, and the real matching objects will not be mistakenly screened. Figure 4b–i visualizes the initial screening results of the eight AHLs, based on the initial screening procedure. Above the dividing line are the sample spectra of the AHLs to be identified, and below the dividing line are the standard spectra of all AHLs in the THz database. After performing the preliminary screening procedure, the standard spectra of AHLs in the database that satisfy the conditions of the preliminary screening (the top three values of $P_{n}$ in Equation (2) with a value of not less than 0.5) will be presented as colored curves, while the spectra that do not satisfy the conditions of the preliminary screening are shown in gray. The visualization results give us a more intuitive understanding of the number of spectra and the standard spectral objects for the preliminary screening.

(2)Comprehensive similarity matching for the identification of different AHLs

After initial screening, the selected spectra were subjected to comprehensive similarity calculations using spectral shape and peak distance features to provide the reliable identification of AHLs with similar molecular structures. Figure 5 depicts the spectral mapping chains and feature peak nodes in the process of matching the target spectra with the spectra that have been screened out from the database. The study utilized the spectral shape feature and peak distance feature to calculate comprehensive similarity, and the target AHLs (black curve) were identified by marking the matching spectrum (color curve) with that possessing the highest comprehensive similarity. The results demonstrate a remarkably high comprehensive similarity between the spectra of the eight types of AHL samples and the corresponding target spectra. The fusion of spectral shape features and spectral distance features is an effective means for accurately identifying the various AHL categories.

The performance of the CSC method was verified by randomly adjusting the weighting parameters α and β. The results indicate that the top matching list can always identify the AHL targets accurately. The quantization accuracy (Acc) value increases with the weight of parameter α, while the degree of differentiation increases with the weight of parameter β. Figure 6 illustrates the spectral similarity matches of AHLs with the parameters α = 0.7 and β = 0.3; the corresponding comprehensive similarity accuracies were sorted in descending order. As shown in Figure 6a–h, all 8 AHL samples (black curves) were exactly identified by comprehensive similarity matching with the top 1 spectrum searched from the database, and the quantified accuracy of the proposed CSC method was over 98.48%. Figure 6i visualizes the top two distributions of Acc values for the comprehensive similarity of all AHLs, further demonstrating the superior Acc values of the top accuracy and the good degree of differentiation with the non-top accuracy. The fusion of multi-scale features of THz spectroscopy successfully achieved the accurate identification of AHLs, providing a novel approach for identifying similar molecular substances. Compared with traditional machine learning, our method is based on the fingerprint property of THz spectroscopy, which can effectively convert molecular information differences into spectral feature differences and, when combined with an efficient comprehensive similarity algorithm, can solve the problem of recognizing similar molecular structures just as effectively. At the same time, our method has fewer data requirements, is a simple process, and the results are more interpretable.

## 3. Materials and Methods

### 3.1. Preparation of Sample Pellets

The standard substances of the eight AHL signal molecules were purchased from Sigma-Aldrich (Sigma-Aldrich Co., St. Louis, MO, USA) and were used without further purification. The basic physicochemical properties of the eight AHLs are listed in Table 2. In the process of sample pellet preparation, the standard substances were first roasted in a dry oven at 50 °C for 4 h, cooled to room temperature, ground thoroughly in an agate mortar, and sieved with 200 mesh. Then, 10 mg of each standard substance was accurately weighed and evenly mixed with 90 mg of polyethylene (PE) powder, respectively (100 mg for each sample). The mixed powders were compressed into pellets of uniform size (diameter: 13 mm; thickness: 1.3~1.8 mm) using a hydraulic press under constant pressure (30 MPa) for 4 min. To minimize the scattering loss, the pellets were produced to be as smooth and parallel as possible. It is essential to note that the thickness of the pellets has no influence on the frequency position of the THz absorption peaks, due to the molecular spectral properties of THz. Therefore, it does not affect the theoretical analysis of the signal molecular fingerprint peaks.

### 3.2. Acquisition of THz Spectra

The spectra of all sample pellets were obtained using the THz-TDS system CCT-1800 (China Communication Technology Co., Ltd., Shenzhen, China), using its transmission-scanning mode. The CCT-1800 has a frequency range of 0.1~4.0 THz and a spectral resolution of 20 GHz. Prior to taking the spectral measurements, all sample pellets were dried to eliminate the interference of moisture on the THz signal. During spectrum scanning, the sample bin was filled with dry nitrogen, and measurements were taken using a stable transmission scan (a spot size of 8 mm and a signal-to-noise ratio of 70 dB at 0.5 THz). To compare the differences between various background matrices, the THz spectrum of dry nitrogen was measured for reference data. The spectrum of the tested samples was obtained as the average of 100 time-domain scans, and each sample was measured 8 times to ensure reliability and accuracy. The 0.5–2 THz band, which has high stability and consistency among the eight measurements, was obtained for subsequent analysis. The acquired THz time-domain signals were Fourier-transformed into frequency-domain signals, then the absorption coefficient spectra of each sample were calculated using Equation (1):(1)$$\alpha_{sam}(\omega )=-10\ln\frac{A_{sam}(\omega )}{A_{ref}(\omega )}$$
where $A_{sam}(\omega )$ and $A_{ref}(\omega )$ are the frequency-domain amplitudes of the sample signal and the reference signal, respectively.

### 3.3. DFT Theoretical Simulations

DFT is a widely used theoretical tool for calculating molecular energies and properties in the fields of physics and chemistry [37]. Electronic structure and energy calculations based on DFT, as well as simulation calculations based on molecular dynamics, have greatly advanced the understanding of microscopic materials [38]. DFT calculations can accurately describe the chemical bonds in molecules, can be used to predict the equilibrium properties of condensed systems, and can also be used to investigate the interatomic forces in molecular dynamics simulations [39]. Among the various functional and basis sets of the DFT, the mixed function B3LYP/6-31G(d,p), which combines the B3LYP function with the 6-31G(d,p) basis set, is widely used to calculate the THz spectra of biomolecules. Therefore, in this work, the theoretical structure optimization and theoretical spectrum simulation computation of eight types of AHLs was performed using the mixed function B3LYP/6-31G(d,p). According to the results of the DFT calculations, by comparing the DFT theoretical spectrum to the THz experimental spectrum, the formation mechanism of the THz fingerprint peaks of samples was revealed [35].

### 3.4. Spectral Similarity Calculations

The THz spectral similarity between eight types of AHL molecules was calculated using the methods of cosine similarity (CS), cosine-cross similarity (CCS), and Jaccard similarity (JS), and the spectral distance was calculated using the methods of Euclidean distance (ED), Hausdorff distance (HD), and discrete Fréchet distance (DFD). It shows that the smaller the distance, the higher the similarity. In terms of spectral shape evaluation, CS calculates the cosine of the angle between two vectors; the closer the cosine is to 1, the more similar these two vectors are [40]. CCS is an improvement of CS in terms of time-series similarity; it calculates the cosine-crossing similarity by comparing the threshold intersection and time lag of time-series data. JS divides the number of intersection elements of two data sets by the number of their concatenated elements; a larger value indicates that the two vectors are more similar [41]. In terms of distance evaluation, the most commonly employed distance algorithms are HD and ED. HD is frequently used to compute the shape difference, which detects the largest deviation between two data sets, whereas ED is used to determine the linear distance between two points. DFD calculates the discrete Fréchet distance between two sets of feature points. The position and order of the feature points along the curve are considered [42]. Herein, we propose a two-step molecular matching identification method for AHLs. In the first step, the AHL standard spectra in the database are initially screened using THz fingerprint peak position character, and the spectra with high peak position matching values are preferentially selected. The first step takes full advantage of the THz spectra fingerprint peak position characteristics to preliminarily screen the spectra and prevent the need for the similarity matching of all spectra in the AHL spectral database. In the second step, a comprehensive similarity algorithm that weights IJS and DFD is used to calculate similarities between the selected spectra and the targets. Considering the possible spectral jitter and offset in the process of spectral detection, it will cause problems in molecular matching recognition. We have improved the JS algorithm to make it less sensitive to slight spectral jitter and shift. The improved Jaccard similarity (IJS) algorithm has a high matching accuracy for spectra of the same molecules, but the overall matching accuracy for spectra of different molecules is also high, and the differentiation is insufficient. The DFD performs well in terms of the distinction degree of similarity accuracy between spectra of the same molecule and the spectra of non-identical molecules. It is possible to achieve high similarity matching accuracy for the same molecules and a high differentiation of matching accuracy for non-identical molecules. Moreover, these are effective methods for evaluating spectral similarity (between two curves) and feature distance (between two sets of feature points), respectively. These methods are simple to implement as they do not require any specific parameters, and the combination of the IJS and the DFD-based weighted algorithm can enhance the accuracy and robustness of the individual algorithm. The flowchart of the AHL matching identification method is shown in Figure 7.

(1) Initial screening based on peak position similarity: According to the fingerprint peak position characteristics of the eight AHLs in the THz spectral database, the spectra with high peak position match the values (the top three values of $P_{n}$ in Equation (2), where the value is not less than 0.5) to the AHL samples that were selected for further spectral similarity calculations:(2)$$P_{n}=\frac{e_{n}}{E_{n}}$$
where *n* (*i* = 1, 2, …, *n*) is the number of spectra in the database to be matched with a target. $P_{n}$ represents the match values of the peak position between the target and the *n*-th standard spectra in the database. $e_{n}$ is the match number of the peak position between the target and the *n*-th standard spectra in the database. $E_{n}$ is the total number of peak positions of the *n*-th standard spectra in the database.

(2) Peak shape similarity matching: For the JS algorithm, the spectral data need to be discretized (the spectra are discretized into *H* (*h* = 1, 2, …, *H*) segments), and all segments need to be first-order differentiated and binary processed (Equation (3)), where 0 and 1 indicate the two attributes of decreasing and increasing spectra, respectively. Meanwhile, if the first-order differentiation equals 0, the attribute value of the previous discrete segment is followed, which facilitates calculation while resolving potential flat-topped peak issues. Therefore, there are four types of value combinations for the marker values of two spectral curves in the same discretized segment $s\left(h\right)$: {0,0}, {0,1}, {1,0}, and {1,1}. Among them, {0,0} indicates that both spectra show a decreasing trend in this discretized segment, while {1,1} indicates that both spectra show an increasing trend. The values {0,1} and {1,0} indicate that the two spectra have opposite increasing and decreasing trends in this discretized segment. However, the vibration and offset in the actual spectra will reduce the accuracy of matching with the standard spectra. To address this issue, a moving window with a width of 10 discrete segments is added. Specifically, for each window, we compared the marker values of their corresponding 10 sets of discrete segments. If more than 9 sets of discrete segments had identical marker values, we considered the inconsistent marker values as outliers and discarded them. Otherwise, all marker values in the window were preserved. The specific process is shown in Figure 8a. By denoting the four cases of {0,0}, {0,1}, {1,0}, and {1,1} as $C_{00}$, $C_{01}$, $C_{10}$, and $C_{11}$, respectively, the IJS between the selected spectra ($s_{i}$) and the target spectra ($s_{t}$) can be calculated using Equation (4):(3)$$\left\{\begin{array}{c} s\left(h\right)=1\;\;s^{\prime }\left(h\right)>0\\ s\left(h\right)=0\;\;s^{\prime }\left(h\right)<0 \end{array}\right.$$
(4)$${IJ}_{i}=\frac{C_{00}+C_{11}}{C_{00}+C_{01}+C_{10}+C_{11}}$$
where *h* is the *h*-th segment in the total number of discretized segments. $s\left(h\right)$ is the marker value at the *h*-th segment. ${IJ}_{i}$ is the IJS value between the spectrum $s_{i}$ and $s_{t}$, respectively.

(3) Comprehensive similarity matching: The IJS algorithm can effectively characterize the spectral shape similarity of two spectra, but it may also produce false high similarity results for parallel spectra that are actually far from amplitude, resulting in matching errors. In contrast, the DFD algorithm (Figure 8b) is commonly understood to be the shortest length required for a leash that will satisfy the condition that the human and the dog each walk the entire track. The DFD algorithm (Equation (5)) can effectively describe the characteristic distance between feature point sets on spectral curves. Therefore, in this paper, the weighted combination of the IJS algorithm (calculating the spectral shape) and the DFD algorithm (calculating the spectral distance) is used to calculate the comprehensive similarity between the selected spectra ($s_{i}$) and the targets ($s_{t}$) from two scales (i.e., spectral shape and spectral distance) in Equation (6):(5)$${DFD}_{i}=min\left\{\underset{j=1,2,\ldots m}{\max}d\left(vs_{i}^{j},vs_{t}^{j},\right)\right\}$$
(6)$$Acc=\alpha \cdot {IJ}_{i}+\beta \cdot \left(1-\frac{{DFD}_{i}}{\sum_{i=1}^{N}{DFD}_{i}}\right)$$
where *N* (*i* = 1, 2, …, *N*) is the number of selected spectra to be matched with a target. ${DFD}_{i}$ is the DFD between the i-th spectrum ($s_{i}$) and the target spectrum ($s_{t}$). *m* (*j* = 1, 2, …, *m*) is the number of features on the spectrum, $d\left(vs_{i}^{j},vs_{t}^{j}\right)$ is the Frechet distance between the features of $vs_{i}^{j}$ and $vs_{t}^{j}$ (on the spectrum of $s_{i}$ and $s_{t},$ respectively). *α* and *β* (*α* + *β* = 1) are the weights of IJS and DFD, respectively. Acc denotes the accuracy of specifying the *i*-th spectrum as the target spectrum.

## 4. Conclusions

This study employed THz-TDS technology to analyze and identify eight types of AHLs with similar molecular structures. The THz fingerprint absorption peaks of the eight AHLs were assigned based on the results of THz experiments and DFT molecular dynamics simulations. A standard THz spectral database was formed, based on the characteristics of fingerprint absorption peaks with C4-HSL (0.92 and 1.58 THz), C6-HSL (0.61, 0.83, 0.97, 1.57, and 1.86 THz), C8-HSL (0.66, 0.87, 1.03, 1.57 and 1.78 THz), 3-oxo-C8-HSL (0.72 and 1.58 THz), C10-HSL (0.73, 1.04 and 1.51 THz), C12-HSL (0.92 and 1.69 THz), 3-oxo-C12-HSL (0.68, 0.87, 1.07, 1.25, 1.46, 1.63 and 1.84 THz), and C14-HSL (0.98, 1.42, 1.61 and 1.75 THz). Based on the THz fingerprint spectral features of AHLs, the spectral database of AHLs was screened using the peak position features of AHL samples, and the similarity between the sample spectra and the selected spectra was determined by combining the spectral shape features and feature point distance features at multiple scales. The similarity results demonstrate that the top matching list can always identify the AHL targets accurately. The Acc value increases with the weight of parameter α, while the degree of differentiation increases with the weight of parameter *β*. By adjusting these weights, a balance between the Acc values of the top accuracy and distinction with the non-top accuracy can be achieved. With *α* = 0.7 and *β* = 0.3, the average quantization accuracy for the eight types of AHL can reach up to 98.48%. Furthermore, our method maximizes the focus on revealing spectral characteristics, thus avoiding the cumbersome processes of traditional machine learning that require massive data sets for training and validation. This work provides the theoretical foundations and empirical evidence for the automatic and rapid identification of AHLs while providing valuable support for investigating the mechanisms of Gram-negative pathogens in QS and QQ-based control strategies for bacterial diseases in aquaculture. In future research, it is recommended to expand the identification categories of AHLs, further optimize the algorithmic framework, and verify the robustness and reliability of AHL identification in natural environments. These factors are crucial for the successful application of this technology in controlling bacterial diseases in aquaculture.

## Figures and Tables

**Figure 1 ijms-25-01901-f001:**
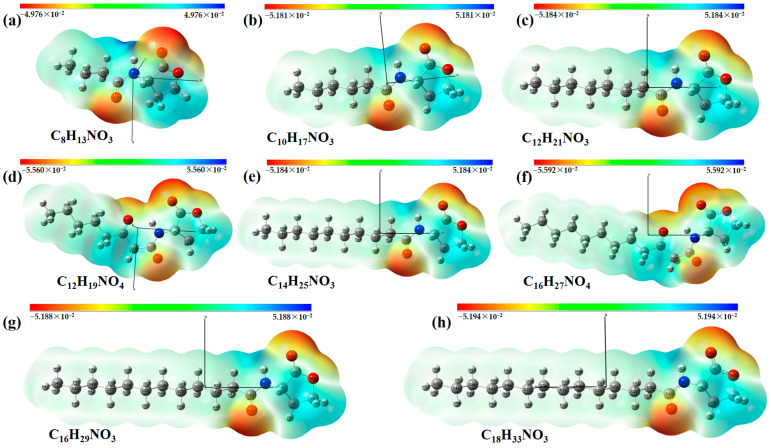
Geometric configuration of AHLs. (**a**) C4-HSL, (**b**) C6-HSL, (**c**) C8-HSL, (**d**) 3-oxo-C8-HSL, (**e**) C10-HSL, (**f**) 3-oxo-C12-HSL, (**g**) C12-HSL, and (**h**) C14-HSL.

**Figure 2 ijms-25-01901-f002:**
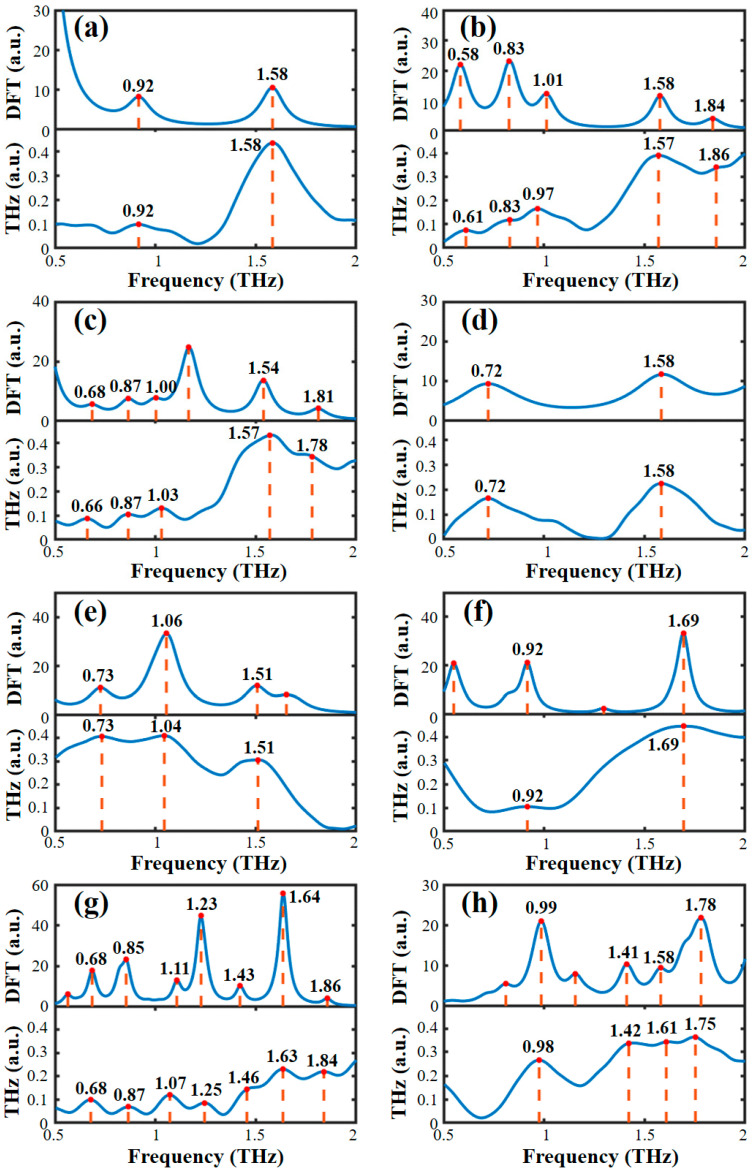
Comparison of the experimental and theoretical spectra of eight AHLs. (**a**) C4-HSL, (**b**) C6-HSL, (**c**) C8-HSL, (**d**) 3-oxo-C8-HSL, (**e**) C10-HSL, (**f**) C12-HSL, (**g**) 3-oxo-C12-HSL, and (**h**) C14-HSL.

**Figure 3 ijms-25-01901-f003:**
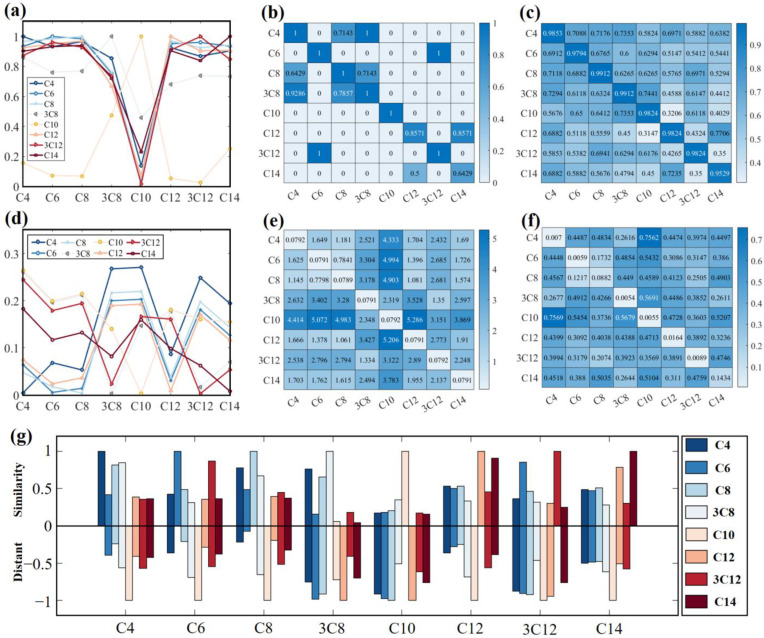
THz spectral similarity calculation for the eight types of AHLs, based on (**a**) CS, (**b**) CCS, (**c**) JS, (**d**) HD, (**e**) ED, (**f**) DFD, and (**g**) the statistical bar chart, based on these six methods. Note: In this figure, C4-HSL, C6-HSL, C8-HSL, 3-oxo-C8-HSL, C10-HSL, C12-HSL, 3-oxo-C12-HSL, and C14-HSL are abbreviated as C4, C6, C8, 3C8, C10, C12, 3C12, and C14, respectively.

**Figure 4 ijms-25-01901-f004:**
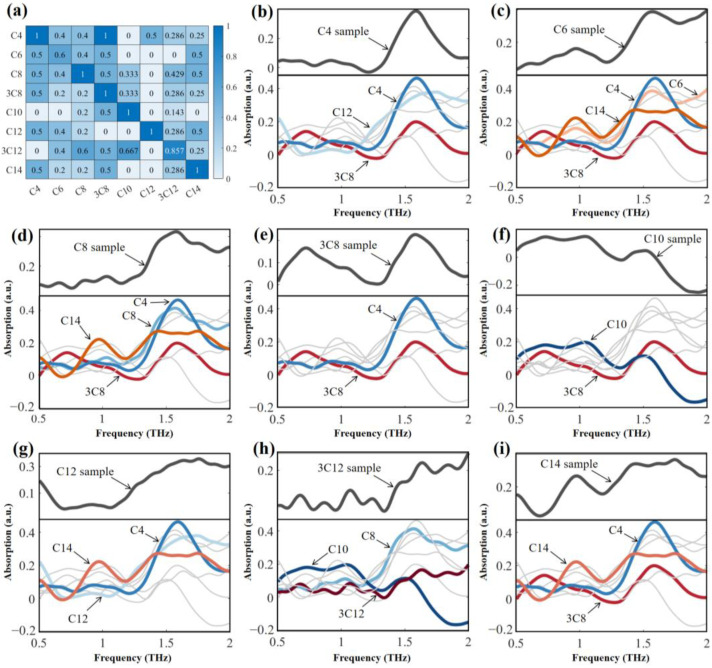
Initial screening based on fingerprint peak position: (**a**) matching values, and the matching results for (**b**) C4-HSL, (**c**) C6-HSL, (**d**) C8-HSL, (**e**) 3-oxo-C8-HSL, (**f**) C10-HSL, (**g**) C12-HSL, (**h**) 3-oxo-C12-HSL, and (**i**) C14-HSL. Note: In this figure, C4-HSL, C6-HSL, C8-HSL, 3-oxo-C8-HSL, C10-HSL, C12-HSL, 3-oxo-C12-HSL, and C14-HSL are abbreviated as C4, C6, C8, 3C8, C10, C12, 3C12, and C14, respectively.

**Figure 5 ijms-25-01901-f005:**
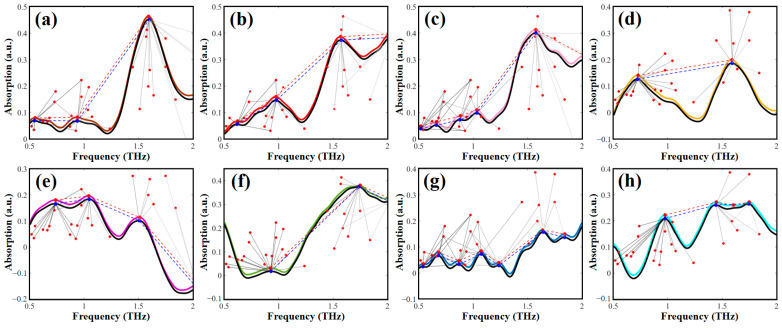
Feature-point searching and matching for (**a**) C4-HSL, (**b**) C6-HSL, (**c**) C8-HSL, (**d**) 3-oxo-C8-HSL, (**e**) C10-HSL, (**f**) C12-HSL, (**g**) 3-oxo-C12-HSL, and (**h**) C14-HSL.

**Figure 6 ijms-25-01901-f006:**
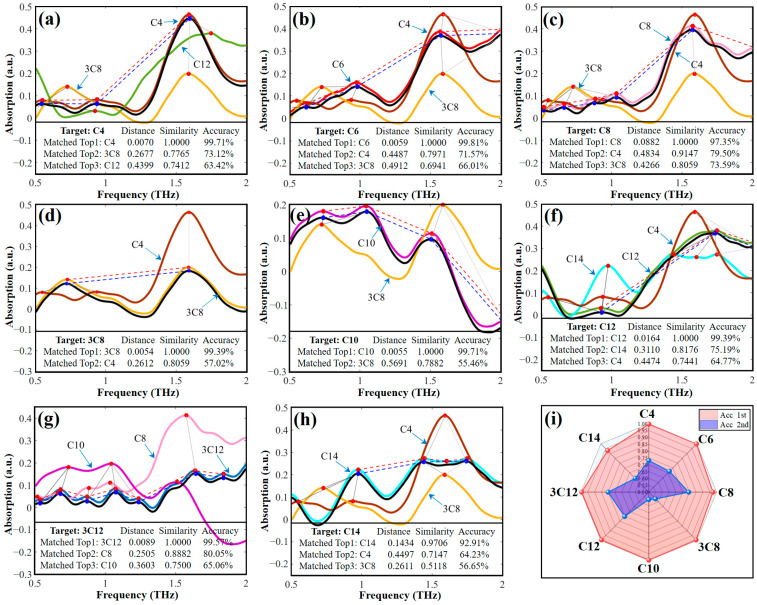
Comprehensive similarity matching to (**a**) C4-HSL, (**b**) C6-HSL, (**c**) C8-HSL, (**d**) 3-oxo-C8-HSL, (**e**) C10-HSL, (**f**) C12-HSL, (**g**) 3-oxo-C12-HSL, (**h**) C14-HSL, and (**i**) Acc visualization. Note: In this figure, C4-HSL, C6-HSL, C8-HSL, 3-oxo-C8-HSL, C10-HSL, C12-HSL, 3-oxo-C12-HSL, and C14-HSL are abbreviated as C4, C6, C8, 3C8, C10, C12, 3C12, and C14, respectively.

**Figure 7 ijms-25-01901-f007:**
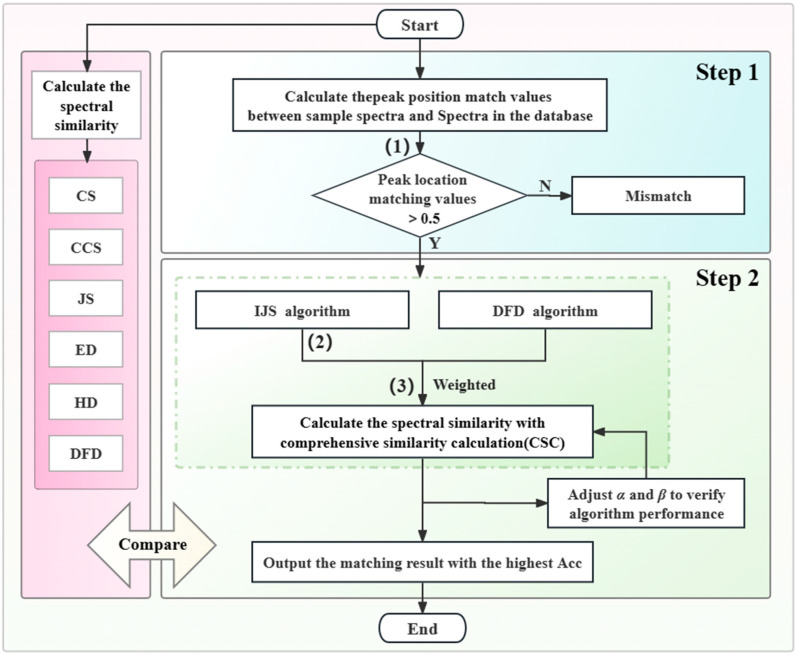
Flowchart of the AHL molecule matching identification method.

**Figure 8 ijms-25-01901-f008:**
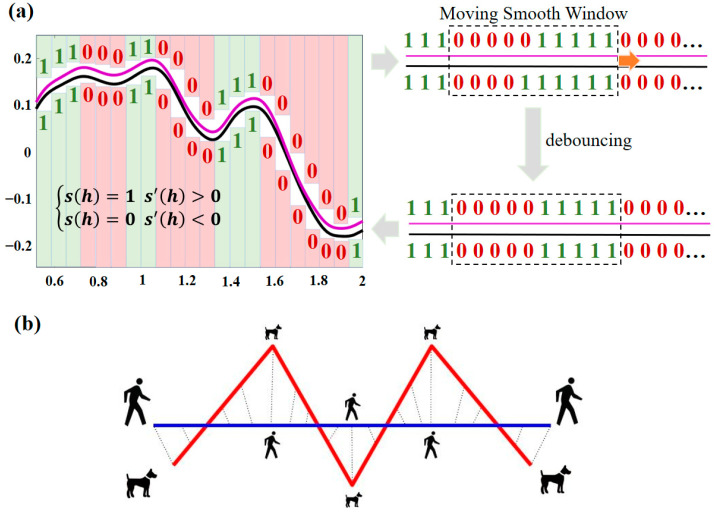
Schematic diagrams of the (**a**) IJS algorithm and (**b**) DFD algorithm.

**Table 1 ijms-25-01901-t001:** Assignment of the absorption peaks.

DFT Peaks (THz)	THz Peaks (THz)	Shift (THz)	Vibration Modes
C4-HSL			
0.92	0.92	0	δ(C-H)oop
1.58	1.58	0	δ(C-N)ip
C6-HSL			
0.58	0.61	0.03	δ(C-H)oop
0.83	0.83	0	δ(C-C)ip
1.01	0.97	−0.04	υbreathing+δ(C-C)ip
1.58	1.57	−0.01	υ(C-C)ip
1.84	1.86	0.02	δ(C-H)ip
C8-HSL			
0.68	0.66	−0.02	υbreathe
0.87	0.87	0	δ(C-C)ip
1.00	1.03	0.03	δ(C-C)ip
1.54	1.57	0.03	δ(C-C)ip
1.81	1.78	−0.03	δ(C-H)opp
3-oxo-C8-HSL			
0.72	0.72	0	δ(C-C)ip
1.58	1.58	0	δ(C-N-C)ip
C10-HSL			
0.73	0.73	0	δ(C-C)ip
1.06	1.04	−0.02	δ(C-C)ip
1.51	1.51	0	δ(C-H)ip+δ(C-C)ip
C12-HSL			
0.92	0.92	0	δ(C=N)ip
1.69	1.69	0	δ(C-H)ip
3-oxo-C12-HSL			
0.68	0.68	0	δ(C-C)ip
0.85	0.87	0.02	υ(C=O)ip+δ(C-C)ip
1.11	1.07	−0.04	δ(C-H)ip
1.23	1.25	0.02	δ(C-N)ip+δ(C-H)ip
1.42	1.46	0.04	δ(C-C)ip
1.64	1.63	−0.01	δ(C-C)ip
1.86	1.84	−0.02	δ(C-H)ip
C14-HSL			
0.99	0.98	−0.01	δ(C-C)ip+δ(C-H)oop
1.41	1.42	0.01	δ(C-C)ip
1.58	1.61	0.03	δ(C=O)ip
1.78	1.75	−0.03	δ(C-C)ip

υ: Stretching vibration; δ: bending vibration; oop: out-plane bending; ip: in-plane bending.

**Table 2 ijms-25-01901-t002:** Physicochemical properties of the AHL signal molecules.

AHL Signal Molecules	CAS Number	MolecularFormula	MolecularMass	MolecularStructure
N-butanoyl-L-homoserine lactone(C4-HSL)	67605-85-0	C_8_H_13_NO_3_	171.19	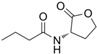
N-hexanoyl-L-homoserine lactone(C6-HSL)	147852-83-3	C_10_H_17_NO_3_	199.25	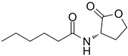
N-octanoyl-L-homoserine lactone(C8-HSL)	147852-84-4	C_12_H_21_NO_3_	227.3	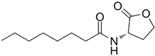
N-(3-oxo octanoyl)-L-homoserine lactone(3-oxo-C8-HSL)	106983-27-1	C_12_H_19_NO_4_	241.28	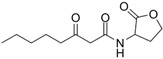
N-decanoyl-L-homoserine lactone(C10-HSL)	177315-87-6	C_14_H_25_NO_3_	255.35	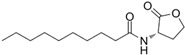
N-dodecanoyl-L-homoserine lactone(C12-HSL)	137173-46-7	C_16_H_29_NO_3_	283.41	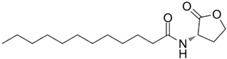
N-(3-oxododecanoyl)-L-homoserine lactone(3-oxo-C12-HSL)	168982-69-2	C_16_H_27_NO_4_	297.39	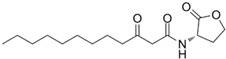
N-tetradecanoyl-L-homoserine lactone(C14-HSL)	202284-87-5	C_18_H_33_NO_3_	311.46	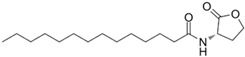

## Data Availability

The raw data supporting the conclusions of this article will be made available by the authors on request.

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
