# Peer review of "Comprehensive Similarity Algorithm and Molecular Dynamics Simulation-Assisted Terahertz Spectroscopy for Intelligent Matching Identification of Quorum Signal Molecules (N-Acyl-Homoserine Lactones)"

_ijms, 2024, doi:10.3390/ijms25031901_

Round 1
Reviewer 1 Report
Comments and Suggestions for Authors
The work presented by Zhang et al. provides an exciting method to determine the presence of quorum-sensing molecules, which could lead to bacteria growth and become a threat to aquatic fauna.
The methodology followed includes a series of calculations and laboratory techniques in order to alert the presence of those compounds.
The use of terahertz spectroscopy results in an interesting and innovative way to determine acyl-homoserine lactones with high accuracy and contributes greatly to the field of microbiology, molecular biology, and even health.
The language is very clear; a couple of observations should be noted:
Line 63-64: Bacteria names are not in italics.
Increase the size of figures 3 and 4.
Author Response
Dear reviewer our response is in the attachment, please see the attachment.

Reviewer 2 Report
Comments and Suggestions for Authors
Manuscript by Lintong Zhang et al. reported a comprehensive similarity algorithm with high accurate for accurate identification of eight Quorum signaling molecules (AHLs) with similar molecular structures based on Terahertz (THz) spectroscopy combined with molecular dynamics simulation and spectral similarity calculation. Firstly, the spectra of eight types of AHLs were measured by THz-TDS system, and the THz fingerprint absorption spectra of eight AHLs were identified, assigned and analyzed by molecular dynamics simulation based on density functional theory (DFT). Then, according to the peak position characteristics of AHLs, the spectrum with high peak matching degree with the target was preliminarily selected to reduce the computational complexity of matching recognition. Next, this paper proposed a comprehensive similarity calculation (CSC) method, which combined the improved Jaccard similarity algorithm (IJS) and the discrete Fréchet distance (DFD) algorithm to calculate the similarity between the selected spectrum and the target and return the matching results with the highest accuracy. The results showed that all AHLs molecular types can be correctly identified, and the average quantitative accuracy of CSC can reach 98.48 %. Overall, this is an interesting contribution that not only solved the problem of quorum sensing mechanism of aquatic pathogens, but also provided a new idea and technical support for intelligent identification of THz spectrum in other fields.
The detailed comments are as follows:
1) The authors didn’t mention the development status of comprehensive similarity algorithm in THz spectroscopy field or molecular recognition field. It is recommended that the author supplement these contents in introduction.
2) At present, are there other comprehensive similarity algorithms in the field of THz spectral recognition for molecules? Compared with the algorithm proposed in this paper, what are the differences in recognition accuracy、algorithm complexity and other aspects?
3) In this paper, the author used Cosine Similarity (CS), Cosine-Cross Similarity (CCS), and Jaccard Similarity (JS) to calculate the spectral similarity, and used Euclidean Distance (ED), Hausdorff Distance (HD), and Discrete Fréchet Distance (DFD) method to calculate the spectral distance, respectively. Could the author briefly describe what is the reason for the combination of IJS and DFD? And what are the advantages of this combination compared with other algorithm combinations?
4) The authors mentioned that each sample was measured 8 times in this paper. What is the measurement error between the 8 measurement results. Will the intensity difference of the same characteristic peak in the 8 measurement results affect the accuracy of similarity recognition?
5) The letter numbers and the format of the horizontal and vertical axes in figure 2 are different from other figures, and the author is recommended to modify them.
Finally, once the above comments are full addressed, the manuscript could be accepted for publication in this journal.
Author Response

(The authors gave the same response as above.)

Reviewer 3 Report
Comments and Suggestions for Authors
This work by Zhang et al. explores the mechanism of Quorum sensing (QS) in aquatic Gram-negative pathogens and explains how the signal is transmitted by putting forth a novel approach for the intelligent matching identification of eight Quorum signaling molecules (AHLs) with comparable molecular structures. A new approach employing terahertz (THz) spectroscopy for the intelligent matching identification of eight Quorum signaling molecules (N-Acyl-homoserine Lac-tones, AHLs) in aquatic Gram-negative bacteria. Spectral similarity computation, density functional theory, and molecular dynamics simulation are all used in the process (DFT). Initial selection of high peak matching value spectra is based on peak location attributes to improve computational efficiency. The work presents a complete similarity calculation (CSC) technique that achieves an exceptional average quantization accuracy of 98.48 percent by combining discrete Fréchet distance algorithms with weighted enhanced Jaccard similarity. In doing so, this study presents a unique, high-throughput, automated technique for aquatic pathogen AHL identification by laying a theoretical and data-supported framework for AHL identification by THz spectroscopy.
A few points the authors could consider in revising their manuscript are listed below:
Major Comments:
1. Understanding the stability criterion is critical in determining the dependability of molecular configurations and subsequent analyses. How were the stable molecular conformations identified, and what criteria were employed to verify that there were no imaginary frequencies? Expanding on the stability evaluation is necessary for evaluating the robustness of the outcomes.
2. How does the structural diversity of the acyl side chain length (4–14 carbon atoms) in the homoserine lactone ring affect the molecular vibration patterns and, therefore, the characteristics of the THz fingerprint? By elucidating the correlation between structural differences and the observed molecular behaviour, the study's findings are rendered more complete.
3. Could you provide more information on the methodology used to assign the vibration modes to the absorption peaks of the AHLs, as shown in Table 1? Enhancing the comprehensibility of the analysis might be achieved by clarification of the approach employed in the assignment of vibration modes. Assuring a transparent and well-documented study, the validity and dependability of the results are contingent upon the procedure by which vibration modes are assigned being crystal clear.
4. In what ways do the knowledge and understandings acquired by the utilization of THz fingerprint peaks in the following identification of AHLs contribute to the molecular dynamics simulation based on DFT? From these results, what are some practical applications or implications that may be deduced? To assess the study's potential influence and applicability in the real world, it is critical to comprehend the practical ramifications of its findings.
5. Could you give additional information about the nature of coupled interactions and how they affect the ensuing THz peaks, such as the benzene ring respiration vibration and the C-C in-plane bending vibration? A better understanding of the molecular dynamics and processes resulting in particular absorption peaks is facilitated by detailed information on coupled interactions. Can you explain why you used a comprehensive similarity method that weighs Improved Jaccard Similarity (IJS) and Discrete Fréchet Distance (DFD)? How did the idea to combine these specific approaches come about, and what benefits do they provide in terms of identifying AHLs? Understanding why a certain mix of similarity algorithms was chosen, as well as the benefits they provide, would aid in evaluating the suggested method's robustness and efficacy. It also sheds light on the factors considered in the algorithm.
6. Could you provide more details on the reasoning behind the fingerprint peak position matching criteria of 0.6 that is indicated in Figure 4? What role does this threshold play in the preliminary screening procedure, and how was it established? Determining the validity of the first screening procedure and the ensuing thorough similarity computations requires an understanding of the threshold-setting criteria.
7. In Figure 6b-i, where sample spectra of AHLs to be recognized are presented alongside standard spectra in the THz database, how were the coloured curves indicating selected spectra picked? What visual or quantitative criteria were employed in their selection? Clarifying the criteria for picking and displaying spectra helps to improve the analysis's transparency and reliability.
8. Traditional machine learning is a laborious technique that necessitates the use of enormous datasets for both training and validation. This is addressed in the conclusion. Could you explain about the distinctions and benefits that this technique to machine learning presents in relation to conventional methods, with particular emphasis on the disclosure of spectral characteristics? A concise comparison or discourse would be appreciated. A deeper appreciation for the ingenuity and effectiveness of the suggested identification technique by examining its comparative benefits in relation to conventional procedures.
9. Are there any specific areas of future research that the report highlights or suggests? What elements of AHL identification or Gram-negative pathogen processes may be investigated further based on these findings? Identifying possible topics for future study assists the scientific community in moving on with more inquiries and advances in the discipline. Answering these questions in the end can give a more thorough and insightful conclusion, making it more useful to readers and scholars in the topic.
Minor Comments:
1. Could you elaborate on the precise procedures and settings applied throughout the B3LYP/6-31G+(d,p) DFT model optimization process? It is essential to comprehend the optimization process in order to evaluate the dependability of the resulting molecular structures.
2. The greatest MEP value is observed in the vicinity of the oxygen atoms in Figure 1. Could you clarify the importance of this discovery with respect to the molecular architectures of AHLs? In particular, the author seeks to appreciate the ramifications of MEP analysis with regard to particular molecular characteristics and their significance to the investigation.
3. What causes or criteria resulted in the elimination of the spectra shown in gray in Figure 6b-i? Understanding why some spectra are eliminated reveals information about the screening process's resilience.
Author Response

(The authors gave the same response as above.)

Round 2
Reviewer 3 Report
Comments and Suggestions for Authors
The authors have put significant effort into addressing my comments and made changes wherever needed. I am happy to recommend this for publication with no further comments.